# Patient–clinician dynamics in remote consultations: a qualitative study of cardiology and rheumatology outpatient clinics in the UK

Elisabeth Grey ,[1,2] Frankie Brown,[1] Paula Smith,[3] Daniella Springett,[1] Dan Augustine,[4] Raj Sengupta,[5] Oliver Peacock,[1] Fiona Gillison[1]

[1]Department for Health, University of Bath, Bath, UK
[2]Bristol Medical School, University of Bristol, Bristol, UK
[3]Department of Psychology, University of Bath, Bath, UK
[4]Royal United Hospitals Bath NHS Foundation Trust, Bath, UK
[5]The Royal National Hospital for Rheumatic Diseases, Bath, UK

**Correspondence to**
Dr Elisabeth Grey;
e.b.grey@bristol.ac.uk

## ABSTRACT

**Objective** Explore the experiences of patients and clinicians in rheumatology and cardiology outpatient clinics during the first year of the COVID-19 pandemic, focusing on the impact of remote consultations on interpersonal dynamics.

**Design** Qualitative study using semistructured interviews, conducted between February and June 2021.

**Setting** The rheumatology and cardiology departments of a general hospital in England, UK.

**Participants** All clinicians and a convenience sample of 100 patients in each department who had taken part in a remote consultation in the past month were invited to take part. Twenty-five interviews were conducted (13 with patients, 12 with clinicians).

**Results** Three themes were developed through the analysis: adapting to the dynamics of remote consultations, impact on the patient's experience and impact on the clinician's experience. The majority of remote consultations experienced by both patients and clinicians had been via telephone. Both clinicians and patients found remote consultations to be more business-like and focused, with the absence of pauses restricting time for reflection. For patients with stable, well-managed conditions, remote consultations were felt to be appropriate and could be more convenient than in-person consultations. However, the loss of visual cues meant some patients felt they could not give a holistic view of their condition and limited clinicians' ability to gather and convey information. Clinicians adjusted their approach by asking more questions, checking understanding more frequently and expressing empathy verbally, but felt patients still shared fewer concerns remotely than in person; a perception with which patients concurred.

**Conclusions** These findings highlight the importance of ensuring, for each patient, that remote care is appropriate. Future research should focus on developing ways to support both clinicians and patients to gather and provide all information necessary during remote consultations, to enhance communication and trust.

## INTRODUCTION

The COVID-19 pandemic precipitated the rapid implementation of remote consultations in both primary and secondary care[1 2]

### STRENGTHS AND LIMITATIONS OF THIS STUDY

⇒ A key strength of this study is considering clinicians' experiences of remote consultations alongside patients', enabling the commonalities and differences between these perspectives to be seen.
⇒ Participants were recruited from two departments of one general hospital—the findings may not be representative of other hospitals.
⇒ This study focused specifically on experiences during the first year of the COVID-19 pandemic—acceptance and expectations for remote consultations is likely to evolve over time as people become more familiar with this format and technology advances.

and remote delivery of health services is set to continue. In the UK, for example, National Health Service (NHS) planning guidance for 2022 sets out the ambition that 'at least 25% (of outpatient appointments) should be delivered remotely by telephone or video consultation (equivalent to c.40% of outpatient appointments that do not involve a procedure)'.[3] Research conducted prior to the COVID-19 pandemic can inform us on user experience when planned remote services are implemented, but the rapid rise in the use of technology during the pandemic means that the background level of skill and acceptability has changed.[4] Recent research has pointed to a need for the postpandemic remote delivery model to evolve in both primary[5] and secondary care[6] to reflect the variation in and changing levels of patients' and clinicians' confidence and skills, as well as changes in technology.

The focus of this article is remote consultations conducted in secondary care (specifically cardiology and rheumatology outpatient appointments). Remote consultations have been used and studied more extensively in primary care[5 7] and, while much of this research will be relevant to secondary care

settings, there is a need to identify the unique challenges and benefits of remote consultations in secondary care too. Research has highlighted some of the advantages and disadvantages of remote consultations for outpatient care during the COVID-19 pandemic. Routinely collected feedback reported advantages, such as reduced stress, enhanced accessibility, cost and time savings,[6] and disadvantages including technical difficulties[5 6 8] and the inability to conduct physical examination.[9–11] Remote consultations are also reported to challenge communication due to the lack of visual cues, such as body language and facial expressions, in telephone consultations (eg, to gauge patient understanding)[4 9 12] and reduced cues (eg, not able to see the patients' movement in and out of a consultation room) and the need to support patients' autonomy while ensuring self-examinations are conducted correctly in the case of videocalls.[4 13]

Relatively limited qualitative research is available exploring some of these difficulties in more depth. Analysis of three studies incorporating interviews with 35 patients with musculoskeletal conditions found acceptability of the consultation format varied independently of characteristics of the condition alone.[4] Clinicians have reported barriers such as a lack of confidence in patients' ability to communicate remotely and patients endorsed clinicians' views that remote consultations are not as satisfactory when building new relationships.[4] Gilbert *et al* reported how interpersonal relations influence the acceptability of video consultations in orthopaedics, suggesting the norms and expectations for remote consultations of both staff and patients play an important role in determining satisfaction with a consultation.

As remote consultations are set to remain a part of health service delivery, it is important to understand how they impact patient–clinician dynamics (ie, the interaction between patient and clinician) and identify ways to ensure they fully meet the needs of both. In this qualitative interview study, we explored patients' and clinicians' experiences of remote consultations conducted during the first year of the COVID-19 pandemic from two outpatient settings in a hospital in south west England, UK, that deliver care for people with chronic conditions: rheumatology and cardiology. This project came about as a collaboration between a University and local hospital interested in exploring how to learn from and improve their delivery of care via telephone and digital technology for the longer term, following initial modes of implementation during the COVID-19 pandemic. Therefore, in this qualitative exploration, our objectives were to understand how patients and clinicians experienced remote consultations and identify factors that facilitated and hindered the provision of remote outpatient care.

## METHODS
### Design
Data were gathered using in-depth semistructured interviews conducted by telephone or online call between February and June 2021. The interview topic guide was developed by the research team, including collaborating clinicians, and informed by a rapid review of published literature on the acceptability, advantages and disadvantages of delivering remote consultations in secondary care (conducted on published research up to February 2021). The interview schedules comprised open-ended questions and prompts to elicit detailed responses, with questions focusing on participants' experiences of face-to-face and remote consultations, markers of good quality remote consultations, and barriers and facilitators to successful remote consultations. Separate interview schedules were developed for patients and staff, and for rheumatology and cardiology, to ensure the question wording was appropriate for participants' experience. Example interview schedules for rheumatology patients and clinicians are provided in online supplemental file 1. No personal data were transferred from the hospital to the research team, and only anonymised interview transcripts were retained after the study, stored securely in accordance with UK data protection regulations.

### Setting
The hospital is a major provider of acute and specialist services in the south west of England. It draws patients from a wide range of socioeconomic backgrounds as well as both rural and urban contexts, but with little ethnic diversity. For this research, departments providing remote outpatient care were invited to take part—four departments responded: dermatology, cardiology, older people's and rheumatology. Work with the dermatology department took a slightly different focus as this team were using a specific software platform. Work with the older people's unit followed a different analysis and is reported elsewhere.[14]

### Participants and recruitment
Clinicians: Emails were sent to all clinicians in patient-facing job roles working in participating departments, inviting them to take part in the study and providing a link to an online information sheet and consent form. The invitation came from clinical colleagues (authors DA and RS, who work at the hospital and collaborated on this project with the other authors, based at the university) but it was made clear that the interviews and analysis would be conducted independently by the research team and participation would be anonymous.

Patients: Patients who had completed at least one remote consultation in participating departments in the past month were eligible to take part. A sample of 100 eligible patients in each of the two departments was identified from patient records and invitation letters and information sheets were sent to them in the post; invitations directed patients to either complete an online consent form or contact the research team directly if they were interested in taking part. To boost recruitment in cardiology, the researcher spent time in the outpatient clinic to explain the study to clinicians and patients to

encourage recruitment. We anticipated a response rate of about 5%–10% from each department; the invitation sample of 100 from each department, therefore, reflects an effort to recruit at least 10 patients.

## Procedure

Participants first completed an online form to provide consent and then demographic information including age group, gender, ethnic group, length of time receiving treatment from the department (patients), and professional role and length of time in profession (clinicians). For some patients, consent and demographic data were collected at the start of the interview calls. Semistructured interviews were conducted remotely, either by telephone or video call, by two researchers (EG and DS).

## Patient and public involvement

Due to the lockdown restrictions in place at the time of study planning and the short time limit in which to complete the study for the funders, we were unable to involve patients and the public in the design and conduct of the study. Findings were, however, reported to patient involvement groups at the hospital and their reflections on these helped to inform a further study.

## Analysis

Interviews were audio recorded and transcribed intelligent verbatim. Personal, identifying information was removed from transcripts prior to analysis. Transcripts were analysed thematically[15 16] using NVivo V.12 software to help organise, code and explore the data. EG and PS (first read a subset of the transcripts to familiarise themselves with the data; EG then coded the subset and compiled an initial coding framework, in discussion with PS. EG and DS (a doctoral student, supervised in analysis by EG) then coded the remainder of the transcripts adding new codes where necessary, continuing to discuss and refine the framework with PS. EG organised the final codes into themes which were discussed and refined with the research team, trying to ensure each theme was distinct and well supported by individual transcripts as well as the dataset as a whole. We took a critical realist perspective, believing that reality exists, independent of observers, but can only be understood through individuals' perceptions and interpretations.[17] We followed an inductive approach to generating codes and themes in the initial analysis, but the selection and development of final themes was driven by the aim of focusing on the psychosocial aspects of consultations. EG and PS are both experienced qualitative researchers with a background in health psychology, DS is a doctoral student who was supervised in the analysis by EG; none work in the hospital. PS was formerly a nurse but did not have experience of remote consultations as either clinician or patient; EG and DS are not clinicians and did not have experience of remote consultations as patients. We have followed the Standards for Reporting Qualitative Research and the completed checklist can be found in online supplemental file 2.

## RESULTS

In total, 25 people responded to the invitations and were interviewed: 12 clinicians (6 cardiology and 6 rheumatology) and 13 patients (3 cardiology and 10 rheumatology). Due to the low numbers involved, overall demographic data for the sample are reported here to protect clinicians' and patients' identities. The clinician participants from each department included both nurses (N=5) and doctors (N=7), seven were male and the majority (75%) were white, with 6–46 years of experience in the profession. All patients were white, the majority (67%) were female and ranged in age group from 35 to 44 years to over 75 years. Patients had been receiving treatment from their respective departments for between 2 months and over 20 years. Most remote consultations conducted in this hospital were by telephone, although a few cardiology clinicians and two patients had also conducted video-call consultations; the results discussed here, thus, mostly relate to telephone consultations except where videocalls are explicitly mentioned. All remote consultations discussed were follow-up appointments as initial consultations were conducted in-person in the hospital. Prior to the pandemic, all consultations (initial and follow-up) took place in-person at the hospital.

Clinicians and patients alike felt the dynamics of remote consultations would be suitable for regular, routine follow-up appointments where the patient's condition was stable; they were considered unsuitable for initial appointments and breaking bad news. Remote consultations were thought to offer advantages in terms of convenience and improving access for patients who live far away from the hospital, who struggle to get time off work for an appointment or who struggle to afford the travel costs. Three themes were developed that went beyond this feedback on the practicalities of remote consultations to explore the interpersonal effects of moving to a remote model of care; the key points for each theme are summarised in Table 1. The first, largest theme, which was consistent across both patient and clinician participants, was adapting to the dynamics of remote consultations. Theme two focused on the Impact on the patient's experience of remote consultations and the final theme was on the Impact on the clinician's experience. The same topics and perspectives were raised by interviewees from both departments, so are presented together within each theme.

### Theme 1: adapting to the dynamics of remote consultations

Clinicians and patients both reported that remote consultations tended to be more focused, with less talking about matters not directly related to the patient's condition.

> Patients don't tend to chat as much either interestingly. People I know who are chatty, and I have to keep on pulling them on track when I'm face to face with them, are much more focused on just what they need to tell me. The actual interview tends to take less time, because they don't side track themselves so

**Table 1** Summary of key points within each theme

| Theme | Key points |
|---|---|
| Adapting to the dynamics of remote consultations | Remote consultations tend to be more focused, 'business-like'—this can make it harder to build a rapport and mean patients do not raise all their concerns<br>The lack of visual cues in telephone consultations can mean it takes longer to gather/convey information.<br>Relying on oral communication to convey empathy, reassurance, encouragement, etc is not always satisfactory.<br>The lack of visual cues in telephone consultations makes it harder to assess patients' understanding. |
| Impact on the patient's experience | The focused style and lack of visual information leaves some patients feeling unable to convey a holistic view of themselves.<br>The physical presence of a clinician can be reassuring; the lack of it can leave patients feeling less well cared for and uncertain of their clinician's level of concern. |
| Impact on the clinician's experience | Relying on only oral communication to gather and convey information can be exhausting.<br>In the majority of cases clinicians are confident of their ability to accurately assess and treat patients remotely, but there is a heightened fear of missing important signs when not seeing patients face to face.<br>The diminished social contact of remote consultations may be less satisfying to clinicians. |

much when they're on the phone. (RC03, rheumatology nurse, female)

This was attributed by some to the different conventions associated with telephone calls compared with those of meeting people in person, with telephone calls being more typically used for short or functional exchanges of information.

> They seem to have been briefer and quicker because we both know the difficulties. I'm kind of used to dealing with technology so I'm used to the way you use it, you talk, they talk … it's a very precise form of communication. (RP12, rheumatology patient, male)

There was some indication that in face-to-face appointments this 'chat' provides a function of establishing rapport and trust and may also contribute to setting a more relaxed tone in which patients feel comfortable raising issues that they worry are less important (explored further in Theme 2).

> There's something about the phone that means that I'm also more business-like and think 'oh no, I won't waste time asking that', and maybe brush over things

a bit more. Whereas if I was sitting in a consultation and actually looking at the consultant, I might open up a little bit more. (CP06, cardiology patient, female)

While this more focused dynamic could work well for patients whose conditions were stable and who had an established relationship with the clinicians, it made it harder to develop a rapport when the clinician and patient had not previously met.

> I would never ever want to meet a person first time over the phone. I think it's always good to look somebody in the eye properly over a table rather than virtually. And then it's a lot easier to have problem phone calls afterwards because you've met them, you know them. You get a sense for somebody, I think. (CP07, cardiology patient, male)

Having to make calls from unknown numbers was also felt to present a barrier to establishing trust—as one doctor highlighted, this required patients to accept that they were not hoax callers.

> I think on a telephone consult it's quite difficult to get across who you are… because our numbers are withheld from the hospital you're taking it for granted that they trust you. Whereas if you've got somebody in a room you build that rapport immediately, I think, by introducing yourself and having eye contact and that kind of thing. (RC01, rheumatology doctor, male)

While remote consultations were perceived to be more focused, clinicians often reported taking longer to question patients as they needed to find new ways of gathering the information needed for treatment decisions that would previously have been evident through visual cues.

> Obviously, you can tell whether somebody looks well or unwell when they walk in… Whereas obviously I don't know that on the telephone, so I have to ask them. You can tell that a bit from the way they talk and what they're talking about, and the energy in their voice, or the lack of energy in their voice. So yeah, there is a little bit of that that you don't see, so you have to ask more questions. (CC04, cardiology doctor, male)

Clinicians discussed how they tried to create space for the discussion of additional questions or concerns in remote consultations, just as they would in face-to-face consultations, to try and encourage greater communication when consulting remotely. They also explained how they tried to use their own body language in face-to-face consultations to express empathy, reassurance or encouragement. While they found means of expressing these emotions orally in remote consultations, this was not always felt to be effective, particularly for 'stoical' patients (ie, those who are reluctant to express that they are in pain).

Because you can use your own body language to actually encourage them to talk, which is really hard to do over the phone; there's only so many ways you can rephrase things until they get fed up with you. (RC03, rheumatology nurse, female)

Sometimes they're not prepared to talk to you over the phone about it. Particularly if you don't know them very well, some people are still reluctant to tell you everything… Some of them are very stoical, our patients. (RC03, rheumatology nurse, female)

The missing visual cues in remote consultations also made it harder for clinicians to assess patients' understanding of the information and advice discussed, again leading them to ask more questions.

I find that I tend to repeat myself quite a lot to check if I'm getting through to them, because I can't assess nodding or smiles or visual clues to know that I'm making sense. So, I constantly say "I hope that's clear", "I hope that makes sense to you". So, it's a bit longwinded I find, some of my consultations, because I keep checking that they've understood what I'm trying to say. (CC01, cardiology doctor, male)

Clinicians also relayed how visual cues from the patient in face-to-face consultations were relied on for determining subtler aspects of their (ie, the clinician's) conduct, such as how to pitch the level of information or their tone of voice.

The way they look at you, whether they look in your eyes or whether they don't look at you at all. That can reflect them being scared a lot of the time, or maybe not understanding. (CC04, cardiology doctor, male)

You can tell how somebody feels by looking at their face and sometimes you can tell if there's something else going on that they want to talk about by looking at them and seeing how they are. You can also tell if somebody's very happy with how they are being treated by their body language. That's been one of the challenges that I've found with telephone consults, is knowing how somebody is just by being around them potentially. (RC01, rheumatology doctor, male)

Despite clinicians tending to ask more questions in remote consultations, the focused dynamic meant there was also a tendency not to leave pauses. Breaks in conversation during consultations are important for allowing patients time to think or to process bad news together and the lack of pauses in remote consultations was felt by both patients and clinicians.

I think it's hard because in a face-to-face consultation you can have pauses, where you're thinking about things… It feels less pressured. Telephone conversations are often 'I talk and then you talk', but there's not much time for a pause or reflection of what has been said. And then it's over, and you can't. It's like often if you finish a consultation, you go to the door

and you sort of say "Oh, what about this, can you just tell me quickly about this before I go?". Whereas with a phone call once it's over, it's over and that's it." (CP09, cardiology patient, female)

If you've just got time just to sit and accept the silence together that's sometimes part of it, and on a telephone you're always going to try and fill that awkward silence. Whereas if they're there with you, you don't have to, you can just let that go and then just wait. (CC04, cardiology doctor, male)

## Theme 2: impact on the patient's experience

When patients were not physically seen by clinicians, some felt that they couldn't convey a full, holistic view of themselves. Patients reported finding it hard to find the right words to discuss their condition, or they could not or did not want to talk about more indirect consequences of their condition or treatment such as weight gain or fatigue. In a face-to-face consultation, they felt the clinician would be able to see these without them needing to raise the topic first themselves

When you're not meeting with them, they're just on the phone, it doesn't feel like they're totally across it. Because they haven't seen you, they haven't examined you. They don't know what you look like… she could probably tell by looking whether I'm in a bad phase or a good phase or whether the bad phases are worse than they used to be. (RP08, rheumatology patient, female)

I think their intuition does play a part in medicine, with doctors… there's nothing quite like sitting face-to-face seeing what colour the skin is, what the general feeling is, what the energy is like. (RP16, rheumatology patient, female)

As one patient pointed out, this may be partially remedied by having video rather than telephone consultations.

Sometimes if it's [a clinician] I don't know they'll say, "do you manage to look after yourself?" … hopefully through a video they can see a bit of my home and how busy life is It gives the impression you feel they are on my wavelength a bit faster. That I don't have a carer coming in!" (RP09, rheumatology patient, female)

The physical presence of a clinician in face-to-face consultations was a clear sign that someone was looking after them and also enabled patients to benefit from the visual cues in a clinician's body language (eg, when expressing empathy or concern).

To see them and be reassured by their demeanour—because you can tell when you look at people whether they're concerned about something or not. They might reassure you verbally, but you can pick up if they think 'oh no this is something we need to investigate further'. (CP09, cardiology patient, female)

Thus, telephone consultations were felt by some to provide less reassurance and this seemed to be felt particularly when patients were suffering worse symptoms or experiencing more uncertainty.

> It would be peace of mind as well, because if the consultant says "I don't think there's any need [for treatment]", at least I feel that they have seen my ailment… if it was me, personally, in pain, I really feel I need to see- I would want to see them face-to-face. (RP15, rheumatology patient, female)

As can be seen from the last quotation, physical presence provides reassurance not only through conveying visual cues of empathy but also through enabling patients to convey what they consider a more holistic picture of themselves, including what they may struggle to express in words.

### Theme 3: impact on the clinician's experience

Clinicians reported that the additional effort needed to gather information, develop rapport and convey interest and understanding solely through speech, without seeing patients, could be exhausting.

> Telephone consults can be quite draining though because you end up talking quite a lot. So, I think if you're doing telephone consults non-stop day in, day out, then from a clinician point of view it would be quite draining. (RC01, rheumatology doctor, male)

> When they have got tender swollen joints it's just really difficult—it's a long drawn out conversation to identify where the problems are if people have got multiple troublesome joints. If it's just a couple that's not really a problem, but if they're having a lot of problems, I just find that a challenging conversation. (RC02, rheumatology nurse, female)

Without being able to see patients or conduct physical examinations, clinicians also reported feeling a fear that they might be missing important signs of disease or misdiagnosing patients. Although in the vast majority of cases clinicians were confident in their ability to accurately assess and treat patients remotely, there could nonetheless be a background-level doubt, particularly for 'stoical' patients.

> There's always going to be that slight niggle at the back of your mind about the patient who thinks they are doing very well but actually they've got some gradual progressive disease that they are not aware of. (RC06, rheumatology doctor, male)

> You will always have a clue, there will be blood tests and the letter … But you do worry, that there might be someone that, because you haven't seen them, that you miss a diagnosis or you don't take on board the severity of a diagnosis. (CC02, cardiology doctor, male)

One clinician also highlighted the potential for safeguarding issues to arise when they couldn't see patients, again causing uncertainty for staff.

> I have had a situation… calling somebody's mobile phone and their partner answering and saying that they were too unwell to come to the phone. And then that brings up like a safeguarding issue—is that patient at risk? If they're so unwell that they can't come to the phone… Are they actually unwell? (RC01, rheumatology doctor, male)

Not being able to see patients meant that clinicians missed an important source of positive feedback, such as patients smiling or acting more at ease following a successful course of treatment. This, in addition to the diminished rapport, made remote consultations less sociable for clinicians.

> It's definitely, in some ways, less satisfying because you can't see the patient smile. And sometimes when patients are walking out the clinic room, you have a bit of banter… There isn't that additional kind of interaction which you get from a face-to-face consult. (CC01, cardiology doctor, male)

> I do think we are visual. I think we do relate to people better when we can see them… I prefer that myself, rather than over the phone. (CC05, cardiology nurse, female)

Overall, the additional sources of uncertainty and diminished rapport and sociability of remote consultations seemed to make them less satisfying for clinicians and potentially more draining. However, the clinicians also acknowledged the practical advantages of remote consultations and some highlighted that with time and experience, conducting remote consultations became easier.

## DISCUSSION

This paper reports on the interpersonal impacts of remote consultations for clinicians and patients in the year following the onset of the COVID-19 pandemic. Both clinicians and patients perceived a shift in the tone of consultations to be more 'business-like' and focused when delivered remotely rather than in person, attributing this to the conventions of telephone calls. Both patients and clinicians flagged the absence of pauses in conversations, removing time for reflection on information delivered, particularly when this was bad or unwelcome news. Clinicians reported having adjusted their approach by asking more questions, checking understanding more frequently and finding ways to express their own understanding and empathy verbally where they would previously have relied on body language. Some found this more tiring than face-to-face consultations. Even with such adaptations, there was acceptance among clinicians that patients seemed less comfortable discussing certain topics and disclosing

as much information as they would have done in person; this may be a particular concern among 'stoic' patients. Supporting clinicians' perceptions, patients reported being less likely to raise 'small' issues in remote consultations that they would have done in face-to-face consultations and felt that they had been less able to communicate a holistic version of themselves.

This study adds to the small body of qualitative research that incorporates both patient and healthcare professional perspectives, which provides an opportunity to explore reciprocal or interacting effects. A qualitative study of patients with musculoskeletal conditions identified that, while remote consultations were suitable and convenient for patients with stable conditions, face-to-face consultations enabled superior communication that was particularly sought when patients were in more distress or pain.[4] These findings were echoed by the patients in our sample, who reported that they would want to be seen in person when their symptoms were bad. Among our sample of cardiology and rheumatology clinicians, we found concerns that patients were not as willing or able to communicate as fully in remote consultations as when face-to-face and an awareness that they do not get a comprehensive picture without seeing patients in person, leaving them uncertain as to whether there is more they should be doing for their patients. Some of the patients and clinicians in our sample speculated that these difficulties may partly be overcome with the use of video call technology (rather than the telephone calls that were employed in most remote consultations). Indeed a survey study of patient and staff perceptions of video consultations conducted from an older adult outpatient unit in the UK found high levels of satisfaction with this format, particularly among patients.[6] In contrast, however, an interview study with patients and clinicians at an orthopaedic centre in the UK found some clinicians perceived video call assessments to be less accurate and more likely to result in worsening of a patients' condition; this corresponded to some patients' experiencing slower rehabilitation and perceiving remote appointments to be less effective than face to face.[18] This may help to explain, in part, the results of a mixed methods study conducted in a UK outpatient department following the rapid implementation of a remote video consultation platform which indicated that, while all patients reported that the consultations met their needs, they were less satisfied than clinicians that they had communicated everything they wanted to (86% vs 95%).[19] A quarter of clinicians showed some recognition of this, reporting a belief that patient experience of a remote video consultation was worse than that in a face-to-face clinic.[18 19] This echoes findings from primary care, where general dissatisfaction with video consultations among both patients and practitioners has been reported, emphasising a belief that the best therapeutic care is delivered in person.[7] Thus, research suggests that patients and clinicians, when communicating remotely, can feel a lack of certainty that either they or the other party has read the situation correctly; this can contribute to perceiving remote consultations to be poorer quality than traditional face-to-face consultations.

## Implications

This study has confirmed that while remote consultations are acceptable and can provide a basis for care, whether delivered in addition to or as an alternative to usual care,[20] they do not provide a like-for-like exchange.[21] Past work has flagged the need to match the format of consultations to different types of appointment (eg, remote consultations are less appropriate for initial appointments and breaking bad news, but may work well for the follow-up of stable cases,[22] but there may also be benefit in matching the format to types of patient (eg, stoic, health anxious, higher risk) and, potentially, to preferences among patients and clinicians.[4] Practical means of identifying how to enable flexible delivery in responding to these variations when implementing at scale may be essential for the successful incorporation of remote consultations into future mainstream care.

Given that some degree of remote care is certain to continue,[21] further work is needed to find effective ways of supporting patients to give, and clinicians to gather, all necessary information during remote consultations. This could help to set the expectations for and, in turn, satisfaction with remote consultations of both patients and clinicians.[18] This may involve, for example, additional training for clinicians in communication techniques for remote consultations, different appointment structures to allow space for reflection, and exploring other approaches that could help to improve both patient and professional confidence and satisfaction. Similarly, educational resources could be developed for patients to provide advice on how they can best prepare and encourage them to mention both major and minor issues and to use their full allocated time slot rather than stick to usual 'businesslike' norms for phone contact.

As both patients and clinicians become more familiar with remote consultations, long-term research will be useful in identifying changes (or the lack of) in the impact of technological restrictions, comfort with communicating via telephone or video call, clinician job satisfaction and other unanticipated consequences of greater remote provision. While patients were generally happy to accept remote care delivered via telephone during the peaks of the pandemic, seeing it as preferable to no care or to increasing their risk of infection by visiting the hospital, this may have changed now that everyday life is increasingly less restricted. Furthermore, the increasing roll-out of remote consultations has important implications in terms of health inequalities, with a greater reliance on technology potentially exacerbating existing disparities between age, ethnic and economic groups as well as for those with disabilities.[4] The small sample of this study prevents us from comparing different patient groups, however, even within our sample there was evidence of some patients being less confident in using digital technology to access healthcare. While providing information

and support to improve digital literacy may help to limit increasing health inequality, the importance of cultural group norms and individual preferences should not be overlooked. By pushing for more consultations to be held remotely, there is a danger that patients who are unable to access or unconfident in using telephones or video call technology may feel they are burdening the system in requesting a face-to-face appointment—this in turn could lead to them not seeking help when they need it, further increasing health inequalities. It is thus important not only for face-to-face consultations to always be available but that they are easily accessible and that patients are made to feel welcome to request such an appointment.

## Limitations

The participants in this study are from a limited pool of respondents from a regional hospital in south west England. Unfortunately, we did not collect data on participants' cultural background or preferred languages, factors which are likely to influence individuals' access to and experience of healthcare.[23] It should also be noted that our sample did not include young adults (<35 years), who are likely to have different acceptance levels and expectations for remote health communication.[24] Therefore, a wider range of considerations and influences may have been discussed if conducted in hospitals based in other communities. While we invited 100 patients from both departments, this was much more successful with rheumatology patients, so these are more strongly represented. Unfortunately, the funding for this research was time limited and so we were not able to conduct any additional recruitment drives to increase or achieve parity in the sample. Most remote consultations within the hospital were conducted via telephone. This is not unusual across the NHS, due to both the availability of equipment and private space for clinicians, as well as concerns about patient access to technology and digital skills. Video consultations are likely to have a different dynamic but were experienced by only two patients in this study. Finally, these interviews took place during a lockdown period and before vaccines were available free of charge for all adults. As such, acceptance of the need to attend remotely was likely higher among both patients and clinicians than when concerns about the risk of infection are lower.

## CONCLUSIONS

The use of remote consultations in secondary outpatient care is largely accepted by patients and clinicians, and set to remain in future healthcare delivery. While technical difficulties in the delivery of remote consultations are rapidly being overcome, more attention is needed on identifying ways to ensure remote consultations are employed appropriately and that, when they are, patients and clinicians are best enabled to communicate effectively. Remote consultations may never be able to fully replicate the interpersonal environment of face-to-face consultations, but research can help to identify ways to ensure that important elements of consulting are not lost and may also highlight traits of remote communication that are particularly suited to certain patients or clinicians.

**Acknowledgements** We would like to thank all patients and clinicians who participated in this study. We are also grateful to the health professionals who provided feedback on the findings in departmental workshops.

**Contributors** EG, FB, FG and PS were responsible for the study conception and design, FG acts as the guarantor. EG and DS conducted the interviews and coded the transcripts; EG was responsible for developing the coding framework and thematic analysis with input from PS. The results and interpretation were discussed among all authors (EG, FB, PS, DS, DA, RS, OP and FG). EG composed a first draft of the manuscript with contributions from FB and FG. All authors commented on the draft manuscript. All authors read and approved the final manuscript.

**Funding** This work was supported by the UKRI Strategic Priorities Fund: Evidence Based Policy Making theme.

**Competing interests** None declared.

**Patient and public involvement** Patients and/or the public were not involved in the design, or conduct, or reporting, or dissemination plans of this research.

**Patient consent for publication** Not applicable.

**Ethics approval** This study involves human participants and was approved by Research Ethical Approval Committee for Health (REACH), Department for Health, University of Bath. Reference Number: EP 20/21 008. Participants gave informed consent to participate in the study before taking part.

**Provenance and peer review** Not commissioned; externally peer reviewed.

**Data availability statement** Data are available on reasonable request. The datasets generated and analysed in this study are not publicly available to reduce the risk of breaking the confidentiality of the study participants, but are available from the University of Bath on reasonable request.

**ORCID iD**
Elisabeth Grey http://orcid.org/0000-0001-9719-9690

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
