## [Reviewer comments · BMJ Open]

ARTICLE DETAILS

TITLE (PROVISIONAL)	Patient-Clinician Dynamics in Remote Consultations: A Qualitative Study of Cardiology and Rheumatology Outpatient Clinics in the UK
AUTHORS	Grey, Elisabeth; Brown, Frankie; Smith, Paula; Springett, Daniella; Augustine, Dan; Sengupta, Raj; Peacock, Oliver; Gillison, Fiona

VERSION 1 – REVIEW

REVIEWER	Wyte-Lake, Tamar Veterans Emergency Management Evaluation Center, USA.
REVIEW RETURNED	27-Dec-2022

GENERAL COMMENTS	I don't think I have ever had no comments to contribute as part of the review process, but at this point, I have no suggestions. The authors have done a wonderful job highlighting the study's key themes, using the illustrative quotes to underscore their findings, identifying their limitations, and connecting their findings to future implications in the field. Thank you for the opportunity to review!
--

REVIEWER	de Camargo Catapan, Soraia The University of Queensland Centre for Online Health, Centre for Health Services Research
REVIEW RETURNED	27-Jan-2023

GENERAL COMMENTS	Thank you for the opportunity to read and review this manuscript. I have made some general comments below that justify the rejection of the article. Title: The title should refer to telephone appointments as remote consultations might include video as well and only 2 participants have experienced video appointments. Plus, all the results and discussions are around the limitations consequent to the lack of visual cues. Article summary - If there are novel insights provided by the study, what are they, and why they are not highlighted in the summary? (line 3, page 4). The only novelty I found in the main text was that “make calls from unknown numbers was also felt to present a barrier to establishing trust” – page 11, line 14.- The recruitment setting cannot be used to justify the limitation in generalizing the study findings (lines 9-10, page 4). Generalization of findings is not the purpose of qualitative research but quantitative (isolating cause and effect, quantifying phenomena, allowing generalisation). For that I suggest some foundation
---

reading such as the introductory chapter of the SAGE Handbook of Qualitative Research (Denzin and Lincoln, 2018).

Introduction

- Line 2-3, page 5: if using a UK reference, it should be clear that this evidence does not apply worldwide – thinking on international readers.

- Assumption without justification or reference, e.g., “but the rapid rise in the use of technology during the pandemic means that the background level of skill and acceptability has changed” (lines 6-9, page 5). As far as I know, the level of acceptability from patient’s perspective hasn’t changed and I didn’t come across studies that report the background level of skill has changed. Happy to be proven wrong.

- Interaction between patients-clinicians been deeply explored during outpatient consultations in the context of the NHS, at the micro level of the study of Prof. Trisha Greenhalgh (Real-World Implementation of Video Outpatient Consultations at Macro, Meso, and Micro Levels: Mixed-Method Study) – these results should be mentioned if the authors want to discuss video appointments.

Methods

- Lines 6-13, page 6: I would recommend the authors to read about different study designs in qualitative research and what questions they respond to. A phenomenological study describes the common meaning for several individuals of their lived experiences of a concept or phenomenon. It aims to reduce individual experiences to a description of the universal essence – a grasp of the very nature of the thing. Another research design would be more suitable to analyse what they called the “impact” that remote consultations have on interpersonal dynamics. Lastly, data reported on in-person consultations is scares and do not allow comparison. Suggested reference: Qualitative Inquiry and Research Design: Choosing among five approaches. Creswell and Poth, 2018.

- Lines 17-20, page 6: the authors mention that a review of literature has been conducted. Has that been published?

- SRQR is attached but not mentioned in the text.

- Line 10, page 7 says interviews and analysis would be conducted independently by the research team and lines 22-23, on the same page, says that interviews were conducted remotely, either by telephone or video call, by two researchers (EG and DS). Do EG and DS work in the same hospital/outpatient department? That should be clarified in the text.

- Line 9, page 7: “The invitation came from clinical colleagues” – are these “colleagues” authors of the study too? If so, they need to be identified.

Results

- Demographic data does not inform if culturally and linguistically diverse (CALD) participants were included in the sample – A population that usually has decreased access to telehealth services and might have different views on it – or how many patients were seen by the first time via remote consultation (not only follow up appointment).

- Most of the results were described as if there was a consensus between all participants about the topics raised. Were there any contrary opinions? Also, the small sample size does not allow us to identify the nuances within the results.

	- In themes 1 and 3 It would be important to differentiate the results from video and telephone as these two modalities have totally different dynamics or if the number of participants who have experienced video consultations was too low, maybe exclude those views from the analysis of this topic (e.g., lines 19-20, page 12; lines 19-21, page 15). The number of participants that experienced video appointments is only mentioned at the end of the study – limitations page 20, line 15. Discussion - Lines 6-7, page 18: to make that affirmation, a systematic review study should be cited. - Structure of the paragraphs in the discussion does not follow the presentation of study findings, comparison with up-to-date scientific literature, and draft conclusions or points for future research. That can lead to mistaken assumptions (e.g., lines 22-25, page 18) - Lines 7-10, page 19: This paragraph could make good use of the definitions of substitutive, alternative, and additive nature of remote consultations appointments (https://pubmed.ncbi.nlm.nih.gov/26343551/) - Lines 21-24, page 19: most interviews were about telephone consultations, but the discussion talks overall about remote consultation. Video and telephone appointments are different and have separate evidence about their efficacy, safety, quality and patient satisfaction, therefore should not be put together risking undermining the benefits of video consultations.
--	---

REVIEWER	González-Rábago, Yolanda Euskal Herriko Unibertsitatea
REVIEW RETURNED	28-Feb-2023

GENERAL COMMENTS	The research topic of the manuscript is interesting and suitable according to the scope of the journal. The understanding of the patient-clinician dynamic in remote consultations is a very necessary research topic to understand the impact in the delivery of care with the increasingly use of remote consultations. However, from my point of view, the manuscript is not ready to be published in the current version, being necessary to revise and explain better the following issues. Before the concrete aspects in each section, I would like to notice the absence of an equity perspective in the design of the study, the explanation of results and the implications for policy. The equity approach, in terms of gender, age and socioeconomic position, should be taken into account when analyzing any topic regarding health and especially in the case of topics such as remote consultations that compromise or hinder access to health care for social groups in vulnerable situations. Introduction Although the content and structure of the introduction is correct, I think that the studied topic deserves a better revision of the scientific literature available until now about the usage of remote consultation in provision of care, including primary care experiences and gathered evidence. The bibliography is focus only on secondary care experiences, which seems to be less developed, but the knowledge about primary care remote consultations should be taken into account. The cleared rephrasing of the research objective at the end of the introduction section would be recommended. Besides, the manuscript does not establish a theoretical hypothesis or an
--

	explanation of the reason of not having an initial assumption about the potential results. Methods It is needed a broader explanation of some issues. Although the checklist of SRQR has been used, there is not enough information about the following topics:  - A mention of the regional hospital of South West England is only done in the Limitations section. It should be clearly explained in Methods why this hospital was chosen, the characteristics of it that may have an impact in the results, and other relevant information that helps the reader to understand the setting and its suitability for carrying out the study. - The reasons under the selection of the department of cardiology and rheumatology is absence. - The explanation about the sampling strategy and the selection of participants should be improved. - In the section Participants and recruitment, for clinicians, it is said that 'invitation came from clinical colleagues': what is the relationship between clinical colleagues and the research team? How was the collaboration requested? - Recruitment of patients: why a sample of 100 eligible patients? What means 'eligible'? Why 100 patients? No explanation about the characteristics of patients that were considered for the selection. Was there any pre-selection according to sex, age, socioeconomic level or other individual or social characteristics that can have an impact in their experiences? - An explanation about the differences between the participants, clinicians and patients, that accept to participate and those who don't. Any difference in the profile that would affect the results? - Number of final interviews carried out: how was set the final number of participants? Why only three cardiology patients? What efforts were done to increase the number of cardiology patients? The number of interviews and reflections about it should be done in methods section. - How was guaranteed the rights of participants regarding the protection of personal data? Results The name and content of the second and third subsections are not clear. 'The impact on the patient' and 'Impact on clinicians' are not self-explanatory. What impact you mean? I think that authors should reflect on what is the contribution of those subsections and the connection with the first one, in order to assess if the identified impacts can be included in the explanation of the adapted dynamics that emerged from the interviews. This would enrich the analysis of each of the dynamics identified. Moreover, some of the explanations should be improved, providing a more analytical descriptions. Some of the ideas are explained in only one sentence and some seems to not have relationship with other, but they have. A mention about 'stoical' patients is done, but it should be better explained. Is this related with the concept of 'difficult patients'? Although it is asked in the interviews according to the interview schedule, the comparison between the remote consultations before COVID pandemic and after/during it is totally absence. It would be necessary to provide some ideas about the perceptions of clinicians and patients about this issue in order to understand the historical context of the results and the influence of it in the adaptation of dynamics.
--	---

	Discussion A deeper discussion of the results with other studies is necessary. It is surprising that any of the articles cited in the Introduction section has been discussed. A reflection on the implications of the implementation of remote consultations in terms of social inequalities in access to healthcare, and therefore in health, is highly recommended.
--	---

VERSION 1 – AUTHOR RESPONSE

Reviewer comment	Response
Please revise the title of your manuscript to include the setting. This is the preferred format of the journal.	We have revised the title to show that the clinics were in the UK (Page 1, line 2 – page and line numbers here refer to the marked copy of the manuscript)
Please revise the 'Strengths and limitations of this study' section of your manuscript (after the abstract). This section should contain up to five short bullet points, no longer than one sentence each, that relate specifically to the methods. The novelty, aims, results or expected impact of the study should not be summarised here.	We have removed the bullet point highlighting the novelty of the study from this section and amended another to highlight a strength of the study. (page 3, lines 8-11)
Please include a copy of the interview/discussion guide (for clinicians and patients) as a supplementary file or a link to where readers can access it.	We have added an example clinician interview schedule to Supplementary file 1, which formerly provided only a patient interview schedule.
Reviewer 1	
I don't think I have ever had no comments to contribute as part of the review process, but at this point, I have no suggestions. The authors have done a wonderful job highlighting the study's key themes, using the illustrative quotes to underscore their findings, identifying their limitations, and connecting their findings to future implications in the field. Thank you for the opportunity to review!	Thank you for this comment, we are very pleased you liked our manuscript!
Reviewer 2	
Title: The title should refer to telephone appointments as remote consultations might include video as well and only 2 participants have experienced video appointments. Plus, all the results and discussions are around the limitations consequent to the lack of visual cues.	As some participants discussed video call consultations, we chose the umbrella term 'remote consultation'. However, we do appreciate that the majority of the findings relate to telephone consultations (as highlighted in the Results and Discussion sections) and so have added a sentence to state this in the Abstract as well (page 2, lines 13-14), which we hope will be useful to readers when searching for relevant literature.
Article summary: If there are novel insights provided by the study, what are they, and why they are not highlighted in the summary? (line 3, page 4). The only novelty I found in the main text was that "make calls from unknown numbers was also felt to present a barrier to establishing trust" - page 11, line 14.	As per the Editor's comments (above) and the author guidance, stating that the purpose of this section is to provide a brief overview of the strengths and limitations of the article, we have tried to focus on strengths and limitations not novel insights in this section. We provide detail of the novel insights in the abstract and, of course, the main text.

The recruitment setting cannot be used to justify the limitation in generalizing the study findings (lines 9-10, page 4). Generalization of findings is not the purpose of qualitative research but quantitative (isolating cause and effect, quantifying phenomena, allowing generalisation). For that I suggest some foundation reading such as the introductory chapter of the SAGE Handbook of Qualitative Research (Denzin and Lincoln, 2018).	We acknowledge that this is an area of qualitative research where there are differing opinions. We take the view of Ritchie et al. (2014) that findings from qualitative research can be generalised but an important aspect of this is clearly stating the conditions under which the research was conducted and describing the sample, in order to consider the extent to which findings may be representative of other situations. We have amended the wording of the Summary bullet point to hopefully clarify this (page 3, line 13).
Introduction: Line 2-3, page 5: if using a UK reference, it should be clear that this evidence does not apply worldwide - thinking on international readers.	We have added a reference to support that the rise of remote consultations in response to the COVID-19 pandemic occurred in many countries throughout the world, of various incomes.
Introduction: Assumption without justification or reference, e.g., "but the rapid rise in the use of technology during the pandemic means that the background level of skill and acceptability has changed" (lines 6-9, page 5). As far as I know, the level of acceptability from patient's perspective hasn't changed and I didn't come across studies that report the background level of skill has changed. Happy to be proven wrong.	We have now provided a reference to support this statement (Paskins et al., 2022) – the reference provides evidence that patients' views on the acceptability of remote consultations has been influenced by a wide range of factors related to the pandemic.
Introduction: Interaction between patients-clinicians been deeply explored during outpatient consultations in the context of the NHS, at the micro level of the study of Prof. Trisha Greenhalgh (Real-World Implementation of Video Outpatient Consultations at Macro, Meso, and Micro Levels: Mixed-Method Study) - these results should be mentioned if the authors want to discuss video appointments.	Thank you for this suggestion. We have now added a reference to the related qualitative analysis by Greenhalgh's group (p4-5, lines 24, 1).
Lines 6-13, page 6: I would recommend the authors to read about different study designs in qualitative research and what questions they respond to. A phenomenological study describes the common meaning for several individuals of their lived experiences of a concept or phenomenon. It aims to reduce individual experiences to a description of the universal essence - a grasp of the very nature of the thing. Another research design would be more suitable to analyse what they called the "impact" that remote consultations have on interpersonal dynamics. Lastly, data reported on in-person consultations is scares and do not allow comparison. Suggested reference: Qualitative Inquiry and Research Design: Choosing among five approaches. Creswell and Poth, 2018.	As the reviewer describes, phenomenology is used to understand individuals' experiences of a phenomenon; in this study the phenomenon of interest was remote outpatient consultations and our analysis looked at how participants experienced these consultations. This manuscript has focused on certain aspects of participants' experience of remote consultation (due to space constraints) but nonetheless we believe the analysis took a phenomenological standpoint. However, we recognise that different opinions on how to define phenomenology exist and so, to avoid confusion, we have removed explicit reference to phenomenology and instead described our ontological perspective, giving further details of our analysis process (page 8-9, 'Analysis')
Lines 17-20, page 6: the authors mention that a review of literature has been conducted. Has that been published?	No, this review has not been published.
SRQR is attached but not mentioned in the text.	Thank you for identifying this! We have now referred to the SRQR checklist in the Methods (p9, lines 6-8) and renamed the attachment 'Supplementary file 2'.

Line 10, page 7 says interviews and analysis would be conducted independently by the research team and lines 22-23, on the same page, says that interviews were conducted remotely, either by telephone or video call, by two researchers (EG and DS). Do EG and DS work in the same hospital/outpatient department? That should be clarified in the text.	EG and DS do not work in the hospital, this has now been clarified in the text (p8, line 24)
Line 9, page 7: "The invitation came from clinical colleagues" - are these "colleagues" authors of the study too? If so, they need to be identified.	Yes, the colleagues were DA and RS, both authors but who work at the hospital rather than the university. We have now stated this on p7, lines 7-8.
Results: Demographic data does not inform if culturally and linguistically diverse (CALD) participants were included in the sample - A population that usually has decreased access to telehealth services and might have different views on it - or how many patients were seen by the first time via remote consultation (not only follow up appointment).	Unfortunately, we only asked participants for their ethnicity and so cannot describe their cultural background or preferred languages. We agree that this is a limitation to our study and have added to the Discussion to highlight this (p23, lines 3-5). We have now clarified on p9, lines 20-21, that all remote consultations were follow-up rather than initial consultations.
Most of the results were described as if there was a consensus between all participants about the topics raised. Were there any contrary opinions? Also, the small sample size does not allow us to identify the nuances within the results.	We did not find directly contradictory opinions among participants but have indicated where not all participants raised certain issues (e.g., by stating that 'some participants felt'). We agree that the small and relatively homogenous sample, while acceptable for qualitative research, is a limitation to our study, as highlighted in the Discussion. However, we would contend that we have identified and reported nuances in the findings (e.g., that the diminished reassurance experienced in remote consultations is more problematic for patients suffering worse symptoms or in times of uncertainty, p16, lines 19-21)
In themes 1 and 3 It would be important to differentiate the results from video and telephone as these two modalities have totally different dynamics or if the number of participants who have experienced video consultations was too low, maybe exclude those views from the analysis of this topic (e.g., lines 19-20, page 12; lines 19-21, page 15). The number of participants that experienced video appointments is only mentioned at the end of the study - limitations page 20, line 15.	We have now added to the beginning of the Results section (p9, lines 19-21) to clarify that, unless explicitly stated, the results relate to telephone consultations. We have chosen to keep in the discussion of video consultations as we feel it provides interesting contrast that the participants highlighted, but hope that the added statement helps to clarify that we do not wish to conflate the two consultation types. We have also added the number of patients that had experienced video consultations to the Results section (p9, line 18).
Discussion: Lines 6-7, page 18: to make that affirmation, a systematic review study should be cited.	This assertion was based on our experience of searching the literature rather than any formal measuring of the number of studies on the patient vs provider experiences. We haven't found a suitable systematic review that reports this and so have removed our statement.
Structure of the paragraphs in the discussion does not follow the presentation of study findings, comparison with up-to-date scientific literature, and draft conclusions or points for	We have now re-structured this paragraph (beginning p19) in the order suggested, which we hope helps to clarify our conclusions.

future research. That can lead to mistaken assumptions (e.g., lines 22-25, page 18)	
Lines 7-10, page 19: This paragraph could make good use of the definitions of substitutive, alternative, and additive nature of remote consultations appointments	Thank you for this suggestion, we have now included reference to this paper (p21, line 7)
Lines 21-24, page 19: most interviews were about telephone consultations, but the discussion talks overall about remote consultation. Video and telephone appointments are different and have separate evidence about their efficacy, safety, quality and patient satisfaction, therefore should not be put together risking undermining the benefits of video consultations.	We have now clarified in this paragraph that our sample mainly discussed telephone consultations (p22, line 7).
Reviewer: 3	
The equity approach, in terms of gender, age and socioeconomic position, should be taken into account when analyzing any topic regarding health and especially in the case of topics such as remote consultations that compromise or hinder access to health care for social groups in vulnerable situations.	We have added a reflection on the implications in terms of health inequalities (p22, lines 9-22). Our sample was small and heterogeneous in terms of age and gender – we are thus wary that it may be misleading to draw attention to differences according to these factors in the Results.
Introduction: Although the content and structure of the introduction is correct, I think that the studied topic deserves a better revision of the scientific literature available until now about the usage of remote consultation in provision of care, including primary care experiences and gathered evidence. The bibliography is focus only on secondary care experiences, which seems to be less developed, but the knowledge about primary care remote consultations should be taken into account.	Some of the papers we draw on in the Introduction and Discussion are actually based on primary care. However, we wanted our article to focus on secondary care, where relatively less research has been conducted. We are mindful that our manuscript is already quite long (as qualitative papers often are) and so wanted to keep the Introduction fairly brief but have now added to this section to clarify our focus on secondary care (p4, lines 13-17).
The cleared rephrasing of the research objective at the end of the introduction section would be recommended. Besides, the manuscript does not establish a theoretical hypothesis or an explanation of the reason of not having an initial assumption about the potential results	Thank you for highlighting this important omission – we have now added an explanation of how the project came about and clarified the research objectives (p5, lines 18-23)
A mention of the regional hospital of South West England is only done in the Limitations section. It should be clearly explained in Methods why this hospital was chosen, the characteristics of it that may have an impact in the results, and other relevant information that helps the reader to understand the setting and its suitability for carrying out the study.	We have now specified that the hospital was in south west England (p5, line 17) in the Introduction section along with an explanation of how the project came about (p5, lines 18-21). We have moved the descriptive details of the hospital from the Limitations section to the Methods in a new subsection, 'Setting' (beginning p6, line 20).
The reasons under the selection of the department of cardiology and rheumatology is absence.	We have now added an explanation of selection in the Methods, Setting subsection.
The explanation about the sampling strategy and the selection of participants should be improved. In the section Participants and recruitment, for clinicians, it is said that 'invitation came from clinical colleagues': what is the relationship between clinical colleagues and the research team? How was the collaboration requested?	As above, we have now added an explanation of how the project came about and clarified the research objectives (p5, lines 18-23). In the Methods, we have also clarified the relationship between the authors (p7, lines 7-8).

Recruitment of patients: why a sample of 100 eligible patients? What means 'eligible'? Why 100 patients? No explanation about the characteristics of patients that were considered for the selection. Was there any pre-selection according to sex, age, socioeconomic level or other individual or social characteristics that can have an impact in their experiences?	In the patients and recruitment subsection, we have now added an explanation for sending invitations to 100 patients from each department (p7, lines 17-19). We also state in this subsection that “Patients who had completed at least one remote consultation in participating departments in the past month were eligible to take part”. We did not conduct any selection, simply taking all people who responded – this is now stated in the Results (p9, line 10)
An explanation about the differences between the participants, clinicians and patients, that accept to participate and those who don't. Any difference in the profile that would affect the results?	Unfortunately, the research team do not have access to demographic data for people who were invited but did not participate in the study. However, our aim was to explore different experiences rather than generate a representative sample.
Number of final interviews carried out: how was set the final number of participants? Why only three cardiology patients? What efforts were done to increase the number of cardiology patients? The number of interviews and reflections about it should be done in methods section.	We have now clarified that we took all people who responded to the invitation (p9, line 10). In our experience it is usual to present the number of participants recruited and interviews conducted in the Results section and so we have kept it there. A member of the research team did try to boost recruitment in the cardiology department by spending time with the clinicians and patients at the clinic to explain the study and encourage participation – this has now been added to the Methods (p7, lines 16-17).
How was guaranteed the rights of participants regarding the protection of personal data?	In the UK, the health research consent process includes providing information to participants about how their data will be used and stored. In this case, personal data that participants provided was stored electronically on secure drives managed by the university. No personal data was transferred between the hospital and university. As part of the ethical approval process, the research team committed to processing and storing participants' data in accordance with UK data protection regulations. We have clarified this in the text p6, lines 17-19.
The name and content of the second and third subsections are not clear. 'The impact on the patient' and 'Impact on clinicians' are not self-explanatory. What impact you mean? I think that authors should reflect on what is the contribution of those subsections and the connection with the first one, in order to assess if the identified impacts can be included in the explanation of the adapted dynamics that emerged from the interviews. This would enrich the analysis of each of the dynamics identified. Moreover, some of the explanations should be improved, providing a more analytical descriptions. Some of the ideas are explained in only one sentence and some seems to not have relationship with other, but they have.	Thank you for this advice, we have gone back over all the themes and made some revisions that we hope help to explain our interpretations of the data. We have also changed the second and third theme titles to 'Impact on the patient's experience' and 'Impact on the clinician's experience', which we hope clarify that it is the patient's and clinician's experience (rather than, say, physical health outcomes) that we are focusing on in these themes.
A mention about 'stoical' patients is done, but it should be better explained. Is this related with the concept of 'difficult patients'?	A clarification of 'stoical' patents has now been added (p13, line 19-20)

Although it is asked in the interviews according to the interview schedule, the comparison between the remote consultations before COVID pandemic and after/during it is totally absence. It would be necessary to provide some ideas about the perceptions of clinicians and patients about this issue in order to understand the historical context of the results and the influence of it in the adaptation of dynamics.	Consultations before the pandemic were all face-to-face in the hospital – we have now stated this at the start of the Results section (p9, lines 21-22). Comparisons between remote and face-to-face consultations are made throughout the Results section.
A deeper discussion of the results with other studies is necessary. It is surprising that any of the articles cited in the Introduction section has been discussed.	We have added to the Discussion to better relate our findings to those of others, including the research cited in the Introduction.
A reflection on the implications of the implementation of remote consultations in terms of social inequalities in access to healthcare, and therefore in health, is highly recommended.	We have now added some reflections on the implications in terms of health inequalities to the Discussion and agree that this is an important issue to highlight in the manuscript. (p22, lines 9-22)

VERSION 2 – REVIEW

REVIEWER	González-Rábago, Yolanda Euskal Herriko Unibertsitatea
REVIEW RETURNED	28-Apr-2023

GENERAL COMMENTS	From my opinion, the manuscript is acceptable for publication.
--